# Dual Voltage Forward Topology for High Efficiency at Universal Mains

**Noam Ezra [1], Toine Werner [2] and Teng Long [1,*]**

1   Department of Engineering, University of Cambridge, Cambridge CB3 0FA, UK; ne285@cam.ac.uk
2   Power Integrations, Cambridge CB4 1YG, UK; toine.werner@gmail.com
*   Correspondence: tl322@cam.ac.uk

**Abstract:** This paper introduces a forward converter aimed at the universal mains voltages, i.e., 220–230 Vac and 115 Vac, named the 'dual voltage forward converter'. The suggested converter has a narrow dynamic range at the universal mains voltages, which results in lower stress on devices, optimal duty cycles, and better overall efficiency. The topology comprises two primary power loops reconfigurable by additional two-state switches and a passive diode, which allows the converter to run in parallel or in series modes and increase the performance over the full universal mains range of 90–265 Vac. The utilization of the devices is better, as they experience lower voltage and current stress by supporting two optimized working points. A converter operating at 100 kHz with an output power of 75 W and output voltage of 12 Vdc was designed and tested. The results were compared with a conventional forward converter with the identical specification. The results at the low mains were similar between the converters; however, at the high mains, the efficiency improvement was between 5% and 23%.

**Keywords:** dual voltage; forward converter; wide input range

## 1. Introduction

Electronic apparatus converting the mains Alternating Current (AC) voltage into a low Direct Current (DC) voltage usually use Switched Mode Power Supply (SMPS) for that purpose. Thus, SMPS can achieve high gain non-inverting output with low voltage stress. In some applications, such as renewable energy and fuel cells, it is possible to use non-isolated topology, such as Single Ended Primary Inductor Converter SEPIC [1]. However, other applications, such as chargers, severs and other systems with human interfaces, need isolation [2]. Flyback and forward topologies are popular due to their simplicity and are common in isolated DC–DC converters for the low power Power Supply Unit (PSU), less than 1 kW. The forward converter high-frequency transformer is not storing energy compared to the Flyback converter, which makes the forward converter more attractive for applications at high output current [2]. However, the main limitation of the forward converter is the limited input voltage range for optimal operation [3–5].

Figure 1 presents a conventional forward converter that supports the universal mains. The rectified input DC voltage from the mains AC voltage is crucial for converter design as it dictates the dynamic range and the voltage/current stress on devices. Below 75 W output power, there is no requirement for power factor correction and a simple passive diode bridge rectifier is usually used to reduce converter cost and complexity [3,4,6,7]. The mains voltage determined the rectified DC voltage, which cannot be controlled without an additional circuit. The world consists of two different ranges of mains voltages, 115 Vac Low Line (LL) used in the USA and Canada, and 220–230 Vac High Line (HL) used in Europe and China [6,7]. To support customer demands and simplify logistics, commercial PSUs supports the full range of mains voltages from 90 Vac to 265 Vac. The support of such a wide range compromises the performance and cost of the converter [3–7]. A universal

mains forward converter need devices that withstand both high voltage stress when used at HL and high current stress when used at LL. That results in challenges in determining suitable devices without compromise of performance and cost.

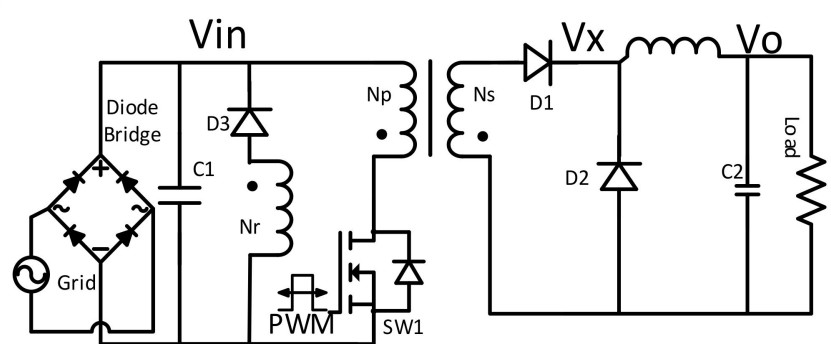

**Figure 1.** Conventional forward converter.

The requirement for better performance, especially efficiency, is growing due to standards and policies, such as Energy Star and 80 plus, which require a minimum efficiency of 86–92% [8]. Furthermore, enhanced efficiency supports a decrease in cooling arrangement, which can reduce the volume and price of the PSU. There are many techniques to increase the efficiency of the conventional forward converter. For high output currents, synchronous rectification is a beneficial technique, which reduces the conduction losses of the output rectifier [9,10]. For high voltage stress, the active clamp [2,11], two-switch forward [4,12] or LC snubber circuit design [4,5,8] are beneficial methods to improve efficiency and, in some cases, even simplify the transformer construction. There are a few suggestions to support efficiency improvement of the forward converter at the universal mains wide range. However, most previous approaches only realize interleaving by mechanically combining several forward converters, connecting two forward converters' primary sides in series to reduce the voltage stress [3,5,13] or in parallel to reduce current stress [8,14]. The main limitations of these solutions are the wide dynamic range over the universal mains and the need for two transformers and two secondary side components. In this paper, the dual voltage forward solution is proposed to address these issues as it has a narrow dynamic range over the full universal mains range, a single transformer with single secondary-side components.

To achieve high efficiency, the converter duty cycle and the inductor ripple current (K = $\Delta$I/2Io) have a significant influence. These parameters are optimized through control and magnetic design to increase efficiency [3]. In most practical applications, the optimal performance (efficiency and size) is achieved at a duty cycle of 50% and a K of 0.1–0.2 [15].

The maximum duty cycle (Dmax) is set by the turns ratio between the primary and reset windings. However, those parameters also influence the voltage stress on the primary Metal Oxide Semiconductor Field Effect Transistor (MOSFET)—Vdsmax. Accordingly, setting low Dmax (Nr > Np) would result in low voltage stress on the primary switch, but a compressed current pulse with higher conduction losses [16]. Nevertheless, setting high Dmax would create the opposite aspects. Another benefit to designing the duty cycle to be about 50% is a better balance between the two rectifiers' (D1 and D2 in Figure 1) conduction losses due to better current stress division [12,15].

K factor is defined as the ripple on the output inductor divided by the load DC current. K linked to the duty cycle and a few other parameters, such as the transformer windings, input and output voltages, the switching frequency, and the inductor value. Optimizing K requires a balance between all the above parameters. Furthermore, a trade-off between the conduction losses of the forward devices (root mean square current) and the inductor size is inevitable.

This paper proposes a new forward converter topology named the Dual Voltage Forward (DVFW). This new topology endeavors to increase efficiency by operating a narrow

dynamic range in both the HL and LL voltages, which enables optimizing parameters like duty cycle and K factor to achieve higher efficiency. Summarized comparisons of operation at the universal mains between the DVFW and the conventional forward converter are presented in Table 1. The operation principle of the DVFW converter is presented in Section 2. A comparison of simulation and hardware results is presented in Section 3, and conclusions of the article are in Section 4.

**Table 1.** Comparison between conventional forward converter and DVFW.

| Parameter | Conventional Converter | DVFW |
|---|---|---|
| Duty Cycle (D) | D at 90 V, 0.25 D at 265 V | D at 90 V, 0.7 D at 265 V |
| Primary switch stress [1] | Vx at 90 V, 4 Vx at 265 V | Vx at 90 V, 1.45 Vx at 265 V |
| Secondary rectifier stress [2] | Vy at 90 V, 4 Vy at 265 V | Vy at 90 V, 1.45 Vy at 265 V |

[1] Vx = 2 Vin assuming Nr = Np, at LL Vx ≈ 200 V and at HL Vx ≈ 750 V. [2] Vy = Vin*Ns/Np neglecting the diode voltage drop.

## 2. Proposed Topology and Operation

Figure 2 presents the proposed DVFW topology. This novel topology extends the efficiency at a broad range of input voltages to support the universal mains. The topology uses better device utilization, enabling the selection of more cost-effective devices. Note that different efficiency enhancement solutions, such as SR, snubbers, active clamp and two-switch forward are compatible with the DVFW and can further increase the overall efficiency. Another advantage of the DVFW is the ability to use an off-the-shelf PWM modulator controller, as the operation is similar to the conventional converter. Note that there is a need to control the delay of one sub-circuit for current sharing purposes, as will be explained later.

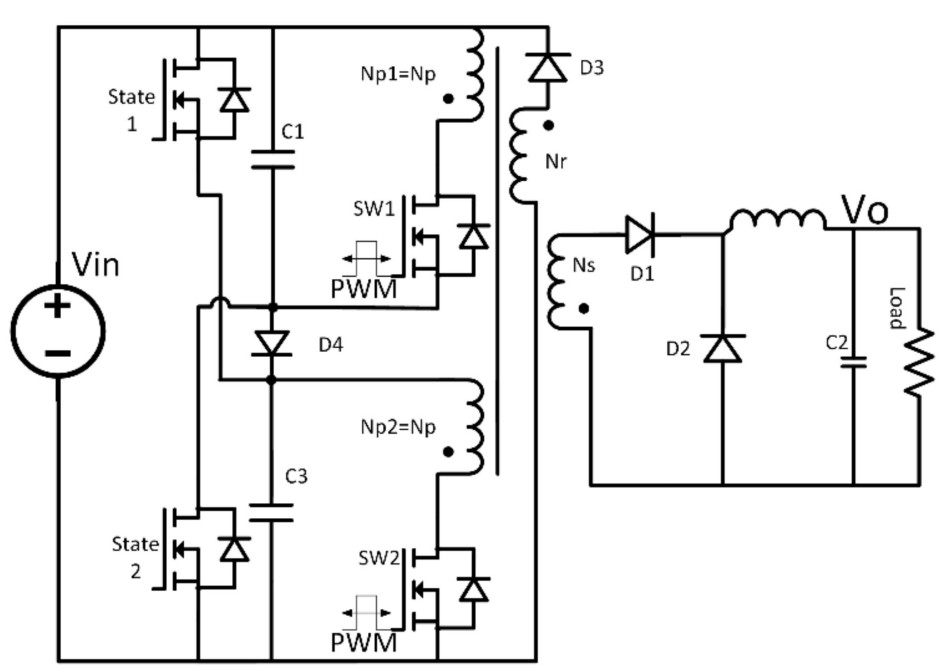

**Figure 2.** Proposed dual voltage forward converter (DVFW).

The DVFW structure is similar to the conventional converter built from two primary sides, which share one four-winding transformer. The reset winding has a similar operation as the conventional forward converter reset winding and discharges the transformer magnetizing inductor. Similarly, the secondary winding has a similar role in both converters to transfer energy to the secondary side. The main benefit of the DVFW is the low voltage rating of the primary side MOSFETs, SW1 and SW2. At LL, they experience similar stress as

the power switch of the conventional forward converter. However, at HL, they experience half the voltage stress of the conventional forward. The transformer is identical with a different pinout ($N_{p1} = N_{p2} = N_p$). The rectified input DC voltage divides between the input capacitors, C1 and C3. To select the operation mode, the state switches, State1 and State2, are employed. At the Low Line Mode (LLM), when the low voltage main is connected (90–130 Vac), the state switches are ON, forcing a parallel connection of the two sub-circuits while the input diode (D4) is in a blocking mode and experiencing LL. At the High Line Mode (HLM), when a high voltage main is connected (180–265 Vac), the state switches are OFF, allowing the two sub-circuits to be connected in series via a diode D4. It is possible to distinguish between the two modes by sensing the input rectified voltage and turning OFF the state switches below a certain threshold, for example, 160 Vac, which will be rectified into 220 Vdc. The state switches will be turned ON about one time at LL and zero at HL. Hence, switching losses can be ignored when selecting state switch MOSFETs, and only conduction resistance (Rds(on)) should be considered. Similarly, the input diode will not experience switching losses.

If the voltage is balanced between the input capacitors (C1 and C3), the state switches and the input diode (D4) experience only low voltage stress. At HLM, each state switch experiences half of the rectified DC input voltage, while at LLM, the input diode experiences the full low rectified DC input voltage. As for the currents, both the input diode and state switches conduct only the ripple current of the capacitors at HLM and LLM, respectively. The diode non-switching nature allows no switching loss, and for the same power rating, the HLM current is nearly half compared to the LLM. Similarly, at LLM, each state switch carries half the input current. Hence, selecting simple low conduction loss devices (input diode and state switches) regardless of the switching energy would keep the losses relatively low.

Figure 3 presents the operation of the conventional forward converter current loops during ON and OFF intervals. As explained in the design considerations, continuous conduction mode (CCM) is preferred; hence, idle mode is not discussed. The charge/discharge of the output capacitor and the reset winding are not presented in the figure because it has a similar operation in both converters.

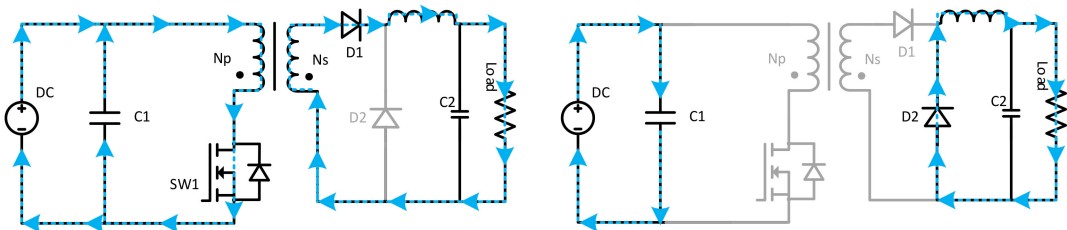

**Figure 3.** Conventional forward converter current loop during primary (**left**) ON time and (**right**) OFF time.

Figure 4 presents the operation of the DVFW at HLM (state switches are OFF). The primary side input DC loop is closed via D4. The input capacitors are connected in series by D4, which conducts the ripple current. The voltage stress on state switches is half of the input voltage, considering voltage balance. Figure 5 presents the operation of the DVFW at LLM (State switches are ON). The primary side input DC loop is closed via D4. The primary side input DC loop divides into two parallel loops, one via State1 and C3 and the second via State2 and C1. The voltage stress on D4 is the rectified DC voltage from the low voltage AC mains.

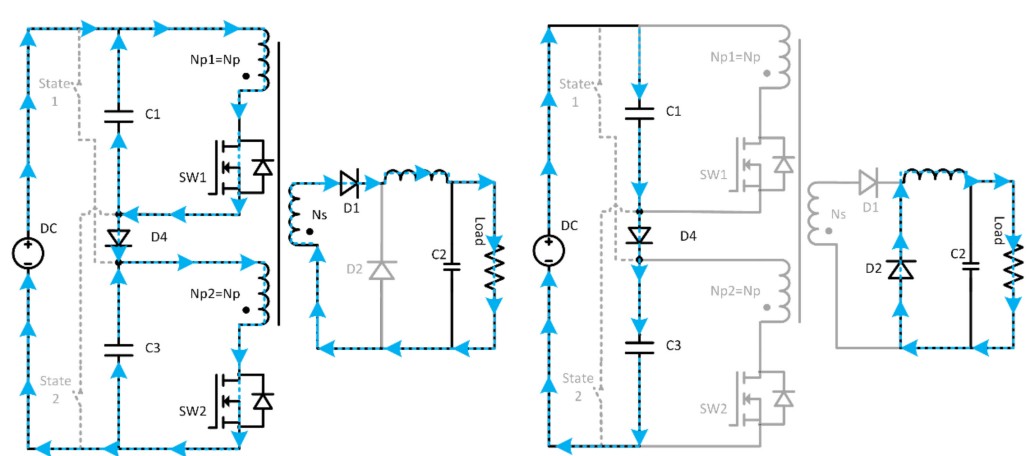

**Figure 4.** DVFW current loop at high line during (**left**) primary ON time and (**right**) OFF time.

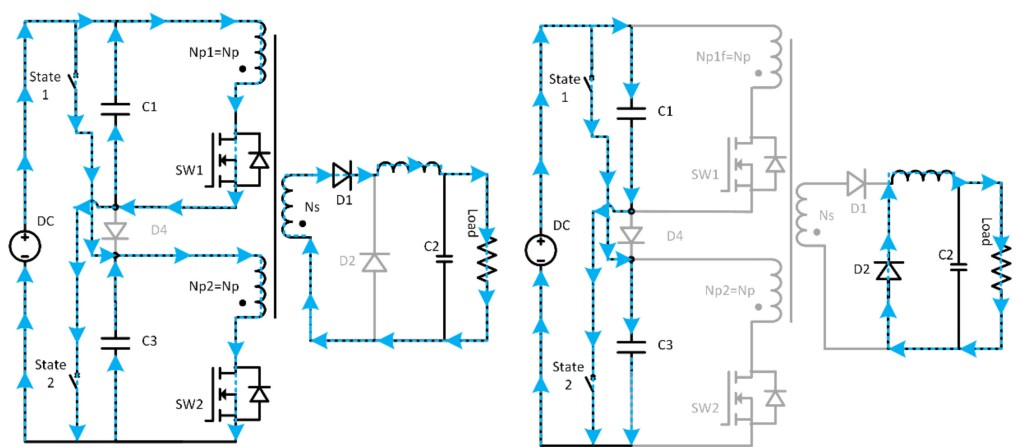

**Figure 5.** DVFW current loop at low line during (**left**) primary ON time and (**right**) OFF time.

The HLM series connection and the LLM parallel connection offers low-stress (voltage and current) requirements from the primary side devices. The secondary side current stress is identical to the conventional forward converter, but the voltage stress on the rectifier is lower due to the DVFW operation. Both modes resemble the conventional forward converter power loops, except that the DVFW use two primary loops instead of one.

## 3. Simulation and Results

### 3.1. Devices Stress Comparison

To compare the topologies performances, first it is necessary to make a device stress comparison, as shown in Table 2. The procedure used for the calculation is identical for the DVFW and the conventional forward converter; however, the DVFW converter experiences a voltage range of 90–132.5 Vac instead 90–265 Vac. That is because the DVFW have two forward converters that will experience 90–132.5 Vac each in both HLM and LLM. Both converters voltage and current stress have been verified by simulation and measurement, which will be presented later in this article. The result shows lower stress experienced by the DVFW devices than the conventional forward converters, which support better devices utilization. For example, in the worst case, the DVFW two primary MOSFET experience 375 V at 0.63 A, compared to the conventional forward converter single MOSFET that experiences 750 V, 1.26 A. Similarly, the DVFW rectifiers experience 67.5 V, 5.6 A and 67.5 V, 3.75 A, compared to 135 V, 5.9 A and 135 V, 3.75 A for the conventional solution. In both converters, the current stress of the D1 rectifier is higher than the D2 rectifier because the duty cycle is lower than 50%. The DRFW duty cycle values at LL is identical

to the conventional forward converter. However, the DRFW duty cycle values at HL are calculated according to the input voltage of Vin/2 because the voltage the series primaries experience is half.

**Table 2.** Parameters comparison between conventional forward converter and DVFW.

| Parameter | Conventional Converter | DVFW |
|---|---|---|
| Duty Cycle | 35% at 90 V [1], **9% at 265 V** [2] | 35% at 90 V [1], **19% at 265 V** [2] |
| Primary switch stress | 200 V@**1.26 A at 90 V** [1], **750 V**@0.65 A at 265 V [2] | 200 V@**1.26 A/2 at 90 V** [1], **375 V**@0.92 A/2 at 265 V [2] |
| Secondary rectifier stress sync/freewheel | 36 V@3.75/5 A at 90 V [1], **135 V@2/5.9 A** at 265 V [2] | 36 V@3.75/5 A at 90 V [1], **67.5 V@2.7/5.6 A** at 265 V [2] |
| Input capacitor | **375 V** at 265 V [2] | **188 V** at 265 V [2] |
| DVFW state switches [3] | **NA** | **0.42**–0.23 A at 90 V [1]–130 V 127–**187 V** at 180–265 V [2] |
| DVFW diodes [4] | **NA** | 100–**185 V** at 90 V [1]–130 V **0.33**–0.22 A at 180–265 V [2] |

[1] Assumed worst-case minimum voltage of 100 Vdc between diode bridge charge cycles (90 V*$\sqrt{2}$)-30 V. [2] Assumed worst-case maximum voltage of 375 Vdc (265 V*$\sqrt{2}$). [3] Voltage stress for state switches at HL only and current stress is LL only. [4] Voltage stress for diode at LL only and current stress is HL only.

### 3.2. Price and Volume Comparison

Although the DVFW requires five additional parts, the devices low-stress feature allows selection of lower-cost devices. Hence, the DVFW can compete with the conventional forward converter. The same product series devices were selected to make a fair comparison between the converters, as shown in Table 3. Note that by taking different product series, it is possible to get even lower-cost devices, which use other technologies.

**Table 3.** Price/size comparison between conventional forward converter and DVFW.

| Parameter | Conventional Converter | DVFW |
|---|---|---|
| Primary switch | 800 V@0.375 Ω STP15N80K5 $t_{d(on)}$ = 19 ns, $t_r$ = 17.6 ns, $t_{d(off)}$ = 44 ns, $t_f$ = 10 ns, TO220 package 1 × £1.69 [1]/0.4 cm$^3$ | 600 V@0.375 Ω STP15N60M2 $t_{d(on)}$ = 11 ns, $t_r$ = 10 ns, $t_{d(off)}$ = 40 ns, $t_f$ = 15 ns, TO220 package 2 X £0.83 [1]/0.4 cm$^3$ |
| Secondary rectifier [2] | 150 V@9.3 mΩ IRFB4115PBF $t_{d(on)}$ = 18 ns, $t_r$ = 73 ns, $t_{d(off)}$ = 41 ns, $t_f$ = 39 ns, TO220 package 2 × 2 × £1.23 [1]/0.4 cm$^3$ | 75@9.3 mΩ IRFB3607PBF $t_{d(on)}$ = 16 ns, $t_r$ =110 ns, $t_{d(off)}$ = 43 ns, $t_f$ = 96 ns, TO220 package 2 × 2 × £0.32 [1]/0.4 cm$^3$ |
| Input capacitor | 400 V@150 uF EKXG401ELL101MMN 1 × £1.63 [1]/40 cm$^3$ | 200 V@150 uF EKXG201ELL151ML25S 2 × £0.86 [1]/20 cm$^3$ |
| Primary driver | 2 A low side gate driver FAN3100 1 × £0.32 [1]/0.008 cm$^3$ | 2 A low side gate driver FAN3100 1 × £0.32 [1]/0.008 cm$^3$ and 4 A high side gate driver FAN37371 1 × £0.48 [1]/0.04 cm$^3$ |
| DVFW current sense | **NA** | Optocoupler 140817142100 2 × £0.14 [1]/0.12 cm$^3$ |
| DVFW state switches | **NA** | 250 V@0.43Ω IPD5N25S3430 2 × £0.29 [1]/0.09 cm$^3$ |
| DVFW diode | **NA** | 200 V@1 A STPS4S200S 1 × £0.16 [1]/0.12 cm$^3$ |
| **Total** | **£8.56/42 cm$^3$** | **£6.48/43 cm$^3$** |

[1] All costs are in https://www.digikey.co.uk (accessed on 29 July 2021) for 1000 units. [2] Used the same MOSFET type for sync and freewheeling rectifier, used two parallel rectifiers due to high current stress.

The DVFW have additional parts, such as a high-side driver that drives the high-side primary MOSFET, an optocoupler that communicates the average current measurement for

the current sharing control loop, state switches, and one input diode for the dual-voltage feature. However, the DVFW low-stress benefit, as presented in the previous section, supports a significant price reduction compared to the conventional forward converter. Typically, a low-stress requirement from a device enables the selection of lower-cost smaller parts. For example, MOSFET with lower breakdown voltage can achieve lower cost and/or better performance (low Rds(on) than a higher breakdown MOSFET; as presented in [17], 400 V MOSFET with lower Rds(on) than 800 V MOSFET is about half the price.

However, due to low-voltage stress, their size decreases dramatically [17], to the point in which the two low voltage input capacitors of the DVFW have a similar size to a single high voltage capacitor of the conventional forward converter. Note that the DVFW additional state switches and diode have very low stress. Therefore, the DVFW capacitors overall addition to the cost and size is small.

It is possible to reduce the DVFW volume and price further by better device selection and integration. In both converters, TO220 packages are used for the primary and secondary MOSFETs. However, due to better efficiency (lower losses at the devices) and spread thermal stress (two primary MOSFET instead of one), the DVFW converter could use smaller packages and less cooling assembly, which supports the further reduction of volume and price. The last opportunity for size and cost optimization is an integrated circuit solution.

The DVFW and the conventional forward converter can use the same transformer with the same amount of copper, i.e., the same length and type of winding wires because the magnetic stress is identical. The only difference is the pinout to accommodate the requirement of the DVFW and conventional forward converter. If the primary winding has an even number of windings connected in parallel, the conventional forward converter will have all these windings connected to the single primary MOSFET, while the DVFW divided them into the two sub-forwards. Therefore, neither the volume nor price of the transformer changed among DVFW and the conventional forward.

### 3.3. DVFW Simulation

The simulation verifies the DVFW features, such as optimal duty cycle, and better devices utilization at both LL and HL. The DVFW has been simulated using LTSpice, along with the conventional forward converter for comparison purposes. The simulation used the devices shown in Table 3 devised by STMicroelectronics, Infineon or LTspice library to achieve an accurate model. The waveforms of the simulation are presented in Figures 13 and 14. The results show that the DVFW can achieve an optimal duty cycle at both the HL and LL, while the conventional forward converter achieves it only at LL and a lower duty cycle at HL as expected.

As for the device stress, the simulation results verify the calculation shown in Table 2. For example, for the DVFW primary MOSFETs, the stress is 375 V at the worst case, while the single conventional forward converter MOSFET experiences 750 V (at HL) and 375 V (at LL). Similarly, the conventional forward converter output rectifiers experience about 136 V (at HL) compared to only 70 V (at HL and LL) in the DVFW topology. The DVFW's additional parts (state switches and input diode) low stress has been verified.

### 3.4. Voltage Balancing and Current Sharing

One of the benefits of the DVFW topology is the low-stress (current and voltage) devices. However, to achieve it, there is a prerequisite to accomplishing balanced voltage between the series-connected sub-forward circuits (MOSFETs and input capacitors) for the HLM and accomplishing balance current between the parallel-connected sub-forward for the LLM.

In LLM, the voltage is balanced because the state switches force parallel connections. In HLM, poor voltage balance would create uneven voltage stress on the devices. Therefore, one of the devices would experience higher voltage stress. The DVFW converter has been simulated with various parameter deviation between the two sub-forward circuits to

discover the impact of device and control parameter tolerance on voltage imbalance. These parameters include gate drivers delay time, transformer primary inductances Lp, input capacitances Cin and MOSFETs resistance Rds(on). Figure 6 shows the simulation results for the voltage misbalance. The results show that the most dominant parameter in the voltage misbalance is the primary inductance, and for every 1% deviation of the primary inductance away from its nominal value $(L_{p1}-L_{p2})/L_{p\text{-nom}}$, a 0.25% imbalance of capacitor voltage occurs $(V_{cap1}-V_{cap2})/V_{nom}$. For example, at 10% inductance deviation (instead 2 mH for both primary inductances, one inductor of 2.1 mH and the other 1.9 mH), the capacitor imbalance would be 5% (instead 375 V divided into two, 196.875 V and 178.125 V obtained in each sub-forward).

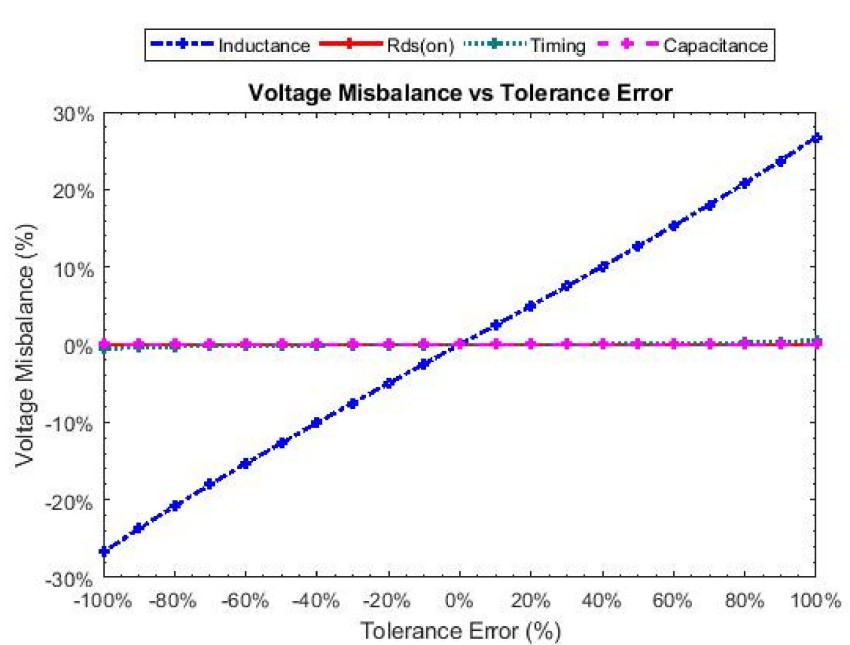

**Figure 6.** DVFW voltage balance vs. primary inductance tolerances at high line.

To explain the mechanism of the voltage imbalance caused by inductance deviation, two inductances, $L_{p1}$ and $L_{p2}$, are assumed for Sub-Forward1 and Sub-Forward2, respectively. When both switches are ON, the reflected voltage that Sub-Forward1 induces on Sub-Forward2 is (1).

$$V_{2-\text{ref}} = V_{LP1}\frac{N_{P2}}{N_{P1}} = V_{LP1}\sqrt{\frac{L_{P2}}{L_{P1}}} \tag{1}$$

Assuming X% deviation between the inductances ($L_{p1}$ is 0.5X% lower than nominal, while $L_{p2}$ is 0.5X% higher) yields (2).

$$V_{LP1} = \frac{V_{2-\text{ref}}}{\sqrt{\frac{L_{nom}(1+0.5X\%)}{L_{nom}(1-0.5X\%)}}} \tag{2}$$

Similarly, analysing the other sub-forward circuit reflected voltage yields (3).

$$V_{LP2} = \frac{V_{1-\text{ref}}}{\sqrt{\frac{L_{nom}(1+0.5X\%)}{L_{nom}(1-0.5X\%)}}} \tag{3}$$

When the switches are ON, the voltage drop on the MOSFETs is negligible. Therefore, the capacitor voltage is approximately the reflected voltage. In consequence, neglecting

the voltage drop on the DVFW diode D4, the reflected voltage sum is the input voltage as shown in (4).

$$V_{in} \approx V_{cap1} + V_{cap2} \approx V_{LP1} + V_{LP2} \tag{4}$$

Combining (2), (3) and (4) and assuming X = 10% yields (5) and (6).

$$V_{cap1} \approx V_{LP1} \approx Vin/2.05 \tag{5}$$

$$V_{cap2} \approx V_{LP2} \approx V_{in}/1.95 \tag{6}$$

Combining (5) and (6) yields (7).

$$V_{blance} = \frac{V_{cap1} - V_{cap2}}{V_{nom}} \approx \frac{\frac{V_{in}}{2.05} - \frac{V_{in}}{1.95}}{\frac{V_{in}}{2}} = 5\% \tag{7}$$

The calculation shows similar results to the simulation, with 5% misbalance when 10% inductance deviation. Therefore, other parameter differences between the sub-forward circuits, such as capacitance, resistance or gate drive delay time, would not significantly affect voltage balancing. The inductance mechanism forces voltage balance through the means of reflected voltage. The sub-forward circuit would act according to the reflected voltage (positive or negative current via the MOSFET) until the voltage is balanced. In real applications with the same number of turns, the difference between the primary inductances is insignificant. Therefore, the voltage imbalance should be negligible, as shown in Figure 14 of the experiment results.

As for current sharing, the critical parameter is the gate driver delay, which depends on the driver IC and the MOSFET source-to-gate capacitance. A simple but effective gate drive circuit for the DVFW can be applied to ensure good current sharing between MOSFETs. This method compares the average current in each sub-circuit and adjusts the delay accordingly, as shown in Figure 7. The additional parts in blue and pink are needed to support the control loop. The delay operation versus the differential amplifier results is explained in the timing diagram. The high side current sense and a fixed delay and the low side current sense (not necessarily isolated) together with the delay set the balancing mechanism. When the error is negative (higher current on the high side), the adjustable gate drive will reduce the low side gate delay and increase the duty cycle of the low side, resulting in a higher current in the low side to have better current sharing. A similar principle applies to the positive error (lower current on the high side). Because the system needs to support sharing in the steady-state, a relatively simple slow control loop is sufficient (much slower than the system control loop). A simple RC filter and an optocoupler or a current sensor followed by an RC filter is fast enough to measure the average current of the MOSFET.

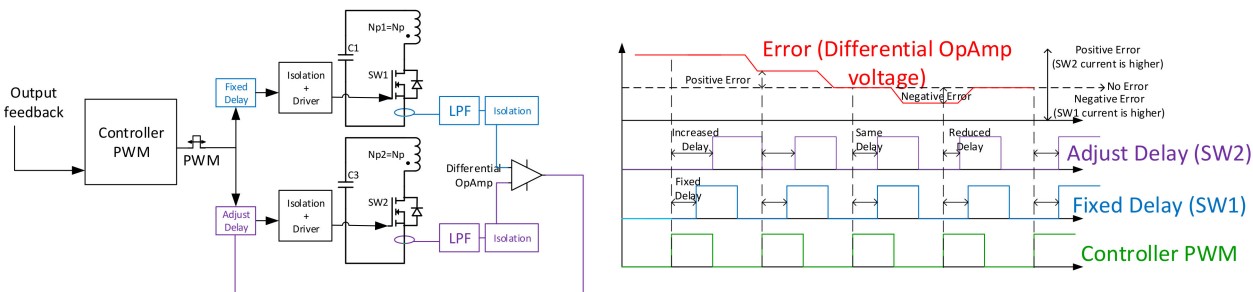

**Figure 7.** DVFW current sharing control loop schematic (**left**) and timing diagram (**right**).

### 3.5. Hardware Setup

The Evaluation Board (EVB) of the DVFW converter and the conventional forward converter has been planned and manufactured for comparison reasons. The models used the devices selected in the price and volume comparison section and presented in Table 3.

The DVFW two low-voltage MOSFETs and two low-voltage capacitors have a comparable volume to the conventional forward converter single high-voltage MOSFET and single high-voltage capacitor, as shown in Figure 8. Furthermore, the DVFW three additional devices (two state MOSFETs and input diode) rated at 200 V to 250 V were assembled on a separate Printed Circuit Board (PCB).

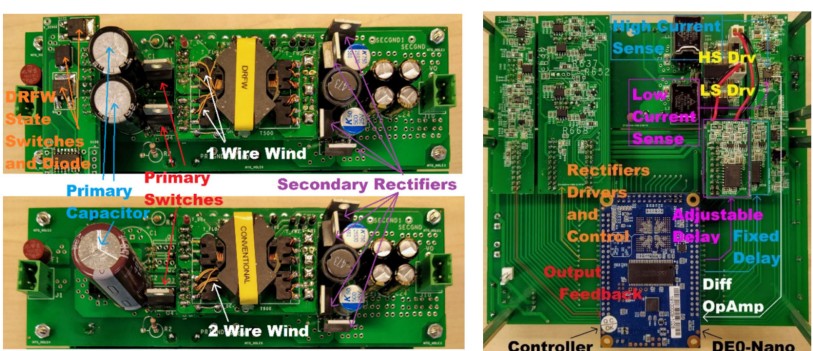

**Figure 8.** Power board of DVFW (**top left**), conventional forward (**bottom left**) and the current sharing control (**right**).

The transformer is a dominant factor of the converter size and cost. The DVFW transformer might need to support more pins, but its overall stress is similar to the conventional transformer converter. Hence, it can have an identical volume and price. Both converters use the same magnetic cores (RM10) and the same amount of copper in the transformers. The only difference is that the conventional forward converter transformer has two primary windings connected to pins 4 and 5, while the DVFW has the windings connected to pins 1 and 2 and pins 4 and 5, as shown in Figure 8. The figure also shows the hardware of the controller for the current sharing control. The current miss-sharing is measured and filtered with an RC circuit, then compared by using differential amplifiers. The comparison result is then sampled by the controller. Using this information, the controller sets the gate driver delay using the delay chips on the circuit.

*3.6. Experimental Results*

This section is divided into three sections: efficiency comparison, current sharing control, and operation comparison.

Efficiency results: The EVB efficiency measurement setup used AC PSU to supply the unit under test (conventional or DVFW converter), the load, and the power meter to measure the unit efficiency as shown in Figure 9. The measurement procedure was setting the target AC voltage and load, simultaneously measuring the input and output energy over a minute, followed by the efficiency calculation of the converter.

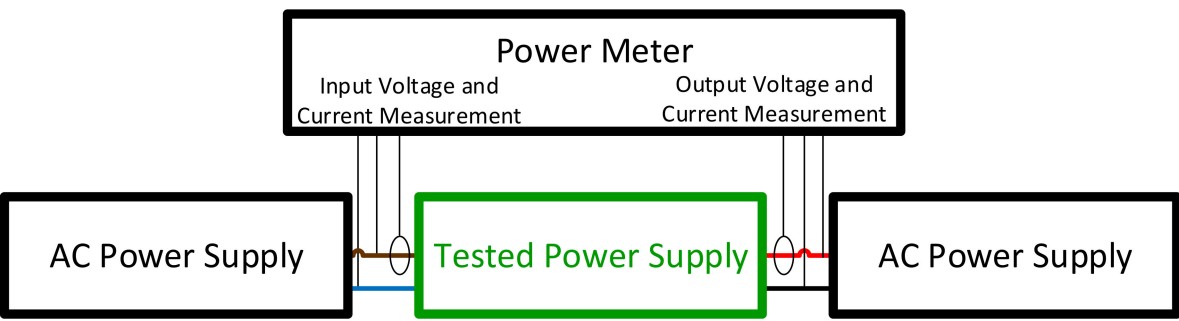

**Figure 9.** The efficiency measurement setup.

The EVB was measured for efficiency at loads from 20% to 100% at the universal mains, minimum voltage (90 Vac), nominal voltages (115 Vac and 230 Vac) and maximum voltage

(265 Vac), as shown in Figure 10. The DVFW achieves similar efficiency at HL and LL. That is due to a similar operating point which consists of parameters such as devices stress, duty cycle and K factor at both mains. Furthermore, at 230 Vac and 115 Vac, when the voltages are exactly double, the efficiency is almost identical because the operating point is alike, as expected. Due to a similar operating point at LL, the conventional forward converter efficiency is similar to the DVFW. However, at HL, the conventional forward converter efficiency is considerably lower at any given load because of the non-optimized nature of the conventional forward converter for universal mains, which cause low duty cycle and high voltage stress as explained above. The higher the voltage, the larger the difference between the converters, with a 5–17% difference at 230 V to 10–23% at 265 V. The results show that the DVFW overall performance at the universal mains is better than the conventional forward converter. The efficiency deviation between the converters demonstrates the DVFW efficiency gain over the full range of the universal mains. The graph takes the DVFW efficiency and substitutes the conventional forward converter efficiency. Therefore, in the sections that the graph is positive, the DVFW converter has better efficiency and vice-versa. Almost all the graph is positive. At HL, the graph is entirely positive and shows a significant efficiency benefit of the DVFW over the conventional forward converter due to the lower stress on devices and optimal duty cycle. At LL, on average, the efficiency of DVFW is only slightly higher than the conventional forward. The LL efficiency results are similar as both converters experience similar stress. However, the main benefit of the DVFW at the HL is due to its low voltage stress on the primary devices, which experience the same stress as DVFW LL operation.

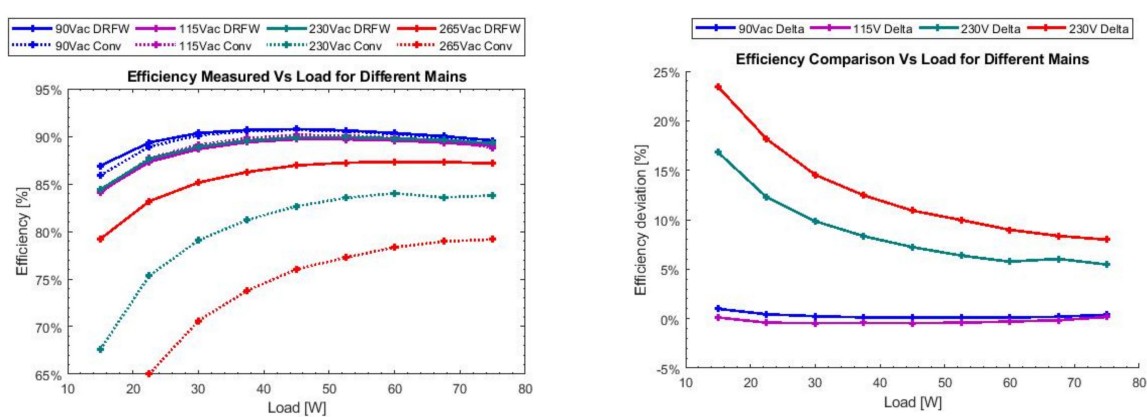

**Figure 10.** DVFW and conventional converter efficiency comparison (**left**) and efficiency deviation (**right**).

Current Sharing Control: To verify the miss-sharing control, the current waveforms were measured. The first step was to verify the current loop operation with two different MOSFETs (STP15N80K5 and STP15N60M2). That emulates extreme tolerance cases, far beyond the tolerances of using the same part numbers MOSFETs, which have similar gate capacitance. The following waveforms were captured in each sub-circuit: the gate signals (green and yellow), the current of the primary winding (purple and orange), the output inductor current (red) and the controller delay in hexadecimal value (blue). The current control loop managed to force these two different MOSFETs to share well, as shown in Figure 11, which shows the gate signals (yellow and green), the MOSFET currents (orange and blue) and the inductor current (red). To further test the control loop, an extreme condition was created by connecting a capacitor of about ten times larger than the gate capacitance in parallel with one of the MOSFETs between the gate and the source. That would change the MOSFET with the capacitor to have a much larger switching delay than the other MOSFET. The captured waveforms are similar to the previous experiment except; the differential amplifier, which measures the current difference, is now presented in pink, and the current of the primary winding is now red and orange (instead of purple

and orange). The results show that the control changed the delay timing to address this extreme miss-sharing condition, and the current is now shared, as shown in Figure 11. The connection of the extra gate capacitance was manual with a through whole capacitor for a short interval to check the control dynamics. At the connection moment, the system had a high miss-sharing current and large current spikes, as shown in the red and orange waveform in the zoomed window 1, and the error (pink) was negative. The controller then adjusted the delay, from 72 to 3C Hex, until the error was small and the currents were balanced with much lower current spikes, as shown in the red and orange waveforms in the zoomed window 2. This extreme test indicates that the current sharing control loop can balance a current even when a large gate drive delay occurs between two sub-circuits.

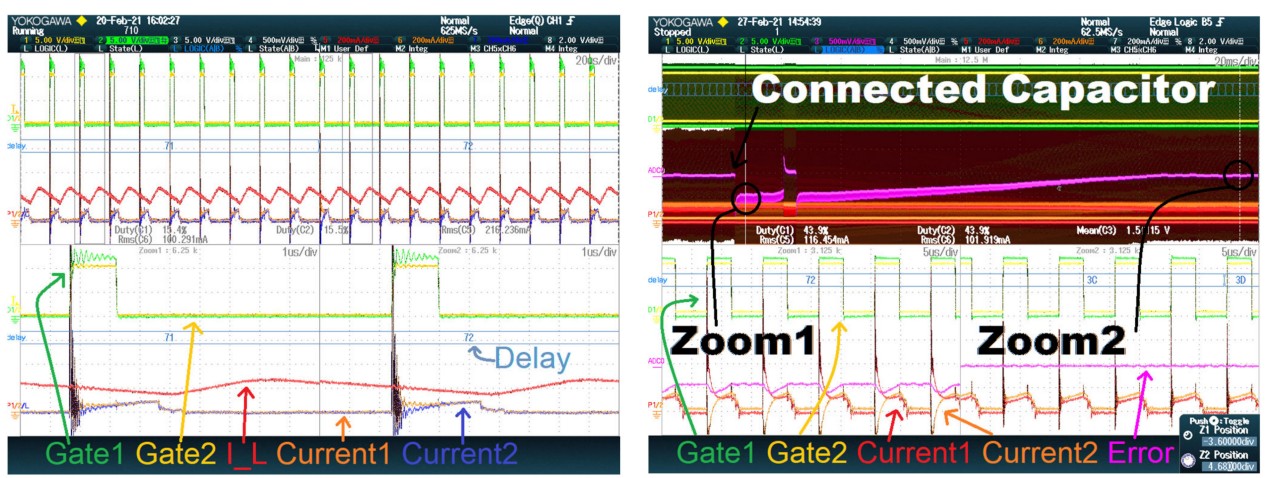

**Figure 11.** DVFW current sharing with different primary MOSFET (**left**), and responding to extreme change in one of the sub-forwards gate capacitance (**right**).

The last experiment illustrates how accurate the closed-loop control is by setting an open-loop with a ±62 ns fixed error. The same waveforms as the previous experiment were captured. Figure 12 shows positive and negative 62 ns timing error. In the positive delay test, the orange current had a smaller delay. Hence, it turned ON earlier and the error was positive. In the negative delay test, the orange current had a larger delay. Hence, it turned ON later and the error was negative. The currents have very large spikes due to the non-sharing effects of the DVFW and the voltage balancing mechanism. This experiment illustrates how sensitive the system is to a small timing error between the gates signals.

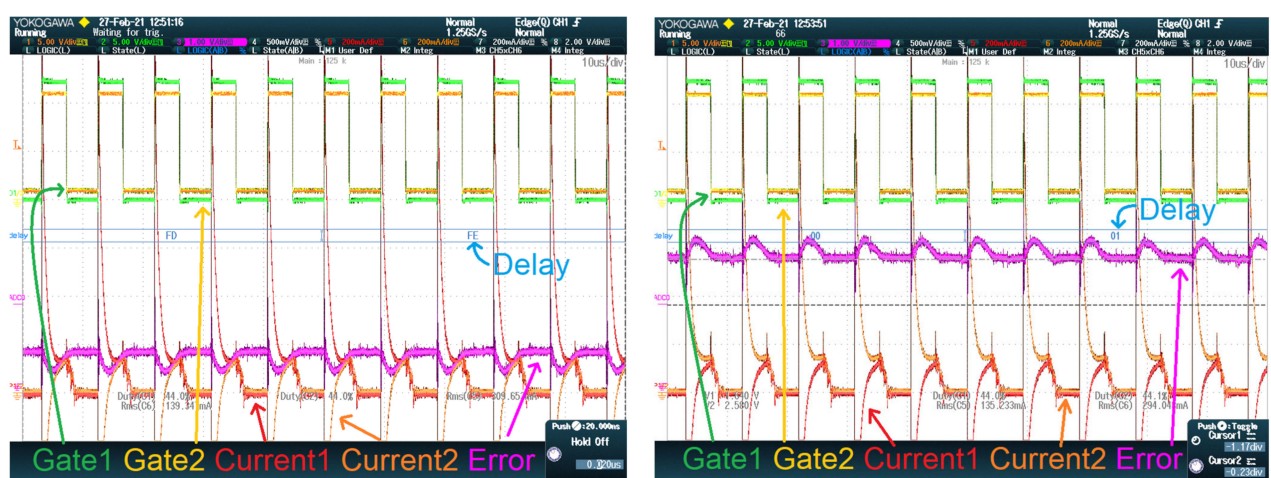

**Figure 12.** Fixed 62 ns timing error between the two sub-forwards: negative error (**left**) and positive error (**right**).

Steady-State results: The steady-state waveforms were measured on the EVB at the same working points as the efficiency measurements; hence, loads from 20% to 100% and mains from 90 Vac to 265 Vac, as presented in Figures 13 and 14. The waveforms measured on the EVB are presented alongside the waveform simulated in LTSpice. The waveforms presented are the full load results at different mains for both topologies. Table 4 contains information about the waveforms, such as the color and description. For simplicity, the waveform color of the simulation and the measurement is identical. The results show that the simulation model is accurate as the measurement results are comparable. For example, the stress on the MOSFETs, rectifiers and capacitors are about the same at all mains. When the operating point is LL, both topology performances are similar, as expected, due to similar stress on devices and working points. However, when the operating point is HL, there is a significant performance difference, as the DVFW devices experience similar stress as the LL operation, and the duty cycle is optimized, as expected. The measurements verified the DVFW balanced capacitor voltage, and the narrow dynamic range as the optimal duty cycle is kept at a wide range of input voltages.

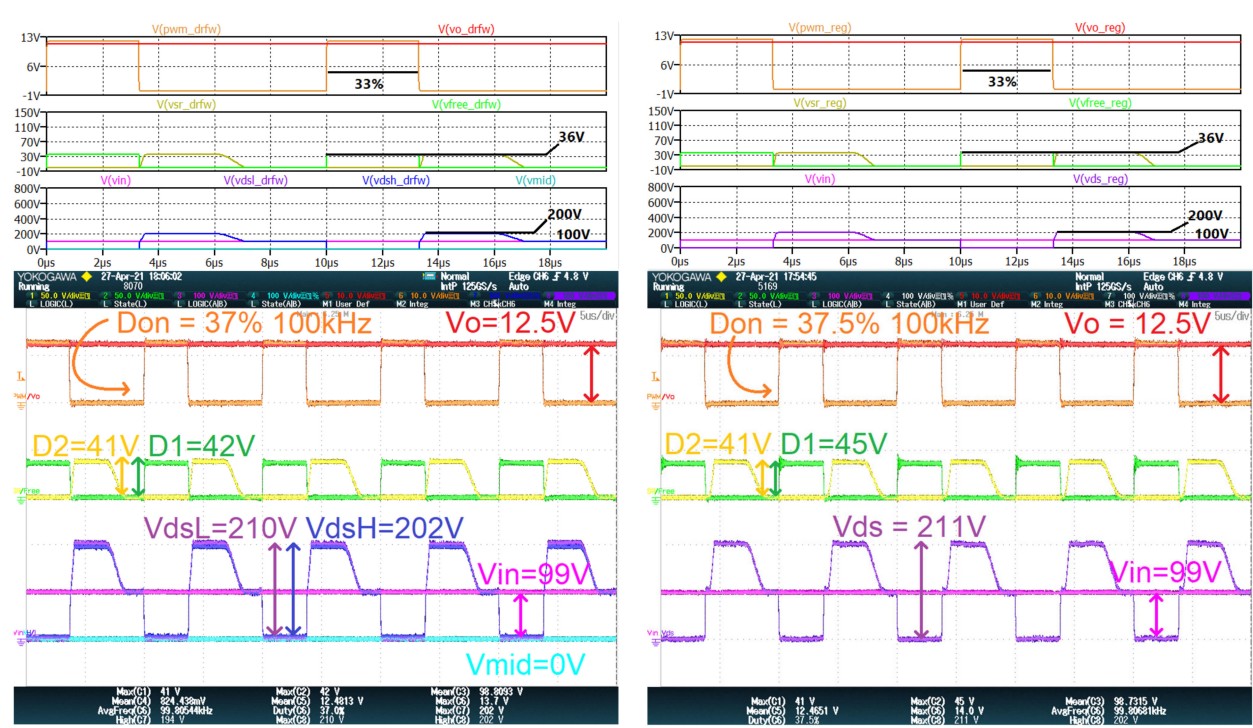

**Figure 13.** Low line minimum voltage (90 V) DVFW simulation (**top left**) and measurement (**bottom left**) vs. conventional forward converter simulation (**top right**) and measurement (**bottom right**).

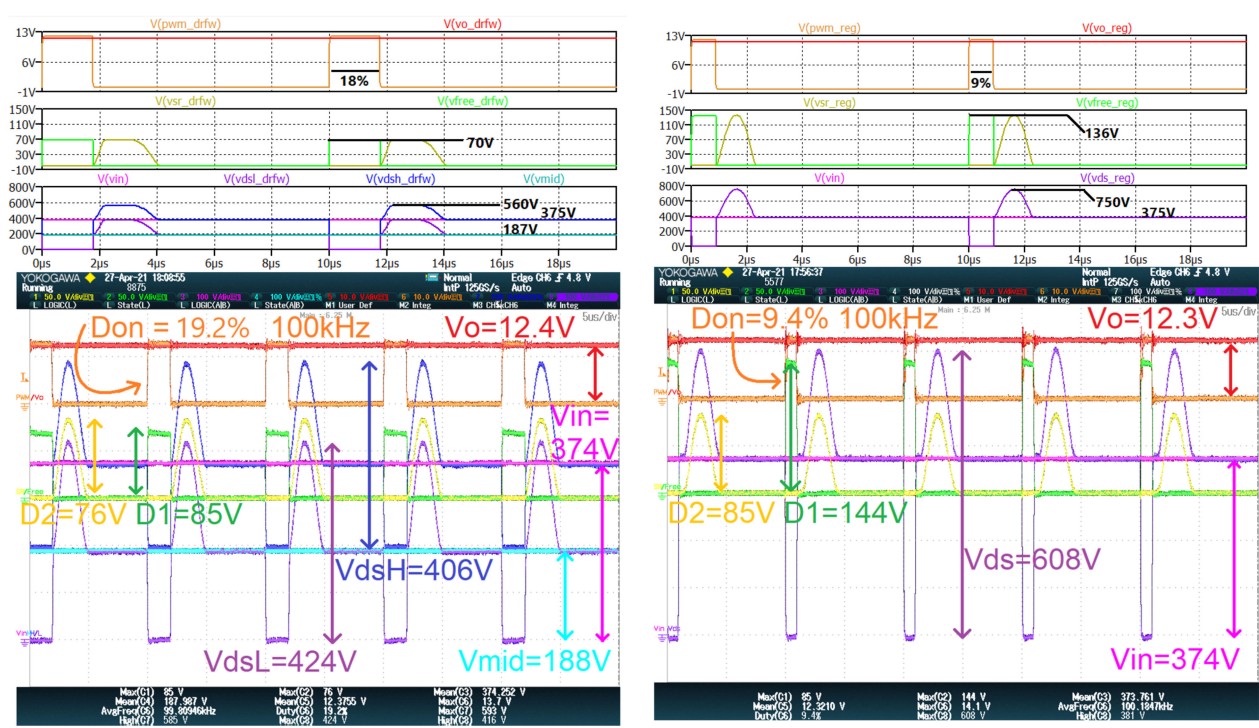

**Figure 14.** High line maximum voltage (265 V) DVFW simulation (**top left**) and measurement (**bottom left**) vs. conventional forward converter simulation (**top right**) and measurement (**bottom right**).

**Table 4.** Figures 13 and 14 waveforms description.

| Signal | Scope Channel | Color | Description |
|---|---|---|---|
| SR Rectifier Voltage | 1 | Yellow | D1 [1] |
| Freewheeling Rectifier Voltage | 2 | Green | D2 [1] |
| Total Input Voltage | 3 | Pink | Vin [1] |
| Voltage on Low Sub-Forward Converter | 4 | Cyan | Voltage between C1 [1], SW1 [1] and D4 [1] |
| Output Voltage | 5 | Red | Vout [1] |
| Duty Cycle | 6 | Orange | Don |
| High Side MOSFET | 7 | Blue | SW1 [1] |
| Low Side MOSFET | 8 | Purple | SW2 [1] |

[1] Symbols are indicated in Figures 1 and 2.

## 4. Conclusions

This paper introduces a new topology built on the conventional forward converter, the DVFW converter. The proposed topology main advantages are better device utilization and narrow dynamic range over the universal mains, which increase the overall efficiency at that range. Both converters have similar performance at the low lines (90 Vac and 115 Vac), and the DVFW performed considerably better at the high lines (230 Vac and 265 Vac), as it achieved better efficiency compared to the conventional forward converter. The DVFW converter can be optimized for both high and low mains, unlike the conventional forward converter that needs to select one of them. The proposed topology enables a narrower dynamic range optimal duty cycle for two different line voltages, such as 115 Vac and 230 Vac, while working with a fixed frequency and using low voltage devices. Although the DVFW part count is larger, its total dimension is comparable to the conventional forward converter, and its total price is lower due to lower rating devices that can be used.

The prototype experimental results verify the calculation and simulation. The comparison between the topologies shows expected benefits of the DVFW, such as lower losses and less stress on the parts than the conventional forward converter. The DVFW could substitute the conventional forward for universal mains, i.e., both the 90–130 Vac and 180–265 Vac. Better device selection and an integrated solution can further optimize the DVFW price and size. The integrated solution can also address the current sharing control loop challenge.

**Author Contributions:** Conceptualization, N.E., T.W., and T.L.; methodology, N.E.; software, N.E.; validation, N.E. and T.L.; formal analysis, N.E.; investigation, N.E.; resources, N.E. and T.L.; data curation, N.E.; writing—original draft preparation, N.E.; writing—review and editing, T.L.; visualization, N.E.; supervision, T.L.; project administration, N.E. All authors have read and agreed to the published version of the manuscript.

**Funding:** This research received no external funding.

**Conflicts of Interest:** The authors declare no conflict of interest.

## Abbreviations

| | |
|---|---|
| SMPS | Switched-Mode Power Supply |
| PSU | Power Supply Unit |
| LL and HL | Low Line (115 Vac) and High Line (220–230 Vac) |
| DRFW | Dual Voltage Forward |
| SR | Synchronous Rectifier |
| LLM and HLM | Low Line Mode and High Line Mode |
| EVB | Evaluation Board |

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
