# Peer review of "Dual Voltage Forward Topology for High Efficiency at Universal Mains"

_electronics, doi:10.3390/electronics11071009_

Round 1
Reviewer 1 Report
- References to figures are not in order into the document. Sometimes is not possible to place them in a good way, but try.
- It would be helpful to have a nomenclature section to easy the reading.
- Sometimes the colours of figures are not the best, like magenta, yellow or green in figures 7, 6, 11, 10, 12 and 13. Try to change when it’s possible.
- Some comments into figures 10, 11, 12 and 13 are too small.
- It will be nice to have a table with data from the devices used in the prototype.
- Could you go deeper in the statement of the efficiency improvement?
- Please, don’t repeat into the conclusions section what you have done, that’s not a conclusion.
Author Response
References to figures are not in order into the document. Sometimes is not possible to place them in a good way, but try. |
Thank you for your comment. To address this issue, Figures 1, 3,4 and 5 moved to be placed closer to their references in the text.
|
It would be helpful to have a nomenclature section to easy the reading. |
Thank you for your suggestion. Added an abbreviated section for easier reading. We have checked the whole document to make sure all abbreviations are defined |
Sometimes the colours of figures are not the best, like magenta, yellow or green in figures 7, 6, 11, 10, 12 and 13. Try to change when it’s possible. Some comments into figures 10, 11, 12 and 13 are too small. |
Thank you very much for these comments. All of the figures mentions (6,7,10,11,12,13) and also figure 9 changed to have no magenta, yellow or light green so it would be more readable for the reader. Furthermore, the text increased in the figures it was too small to read for better visibility. Note that a new figure was added to the article (Fig. 9) which pushed figures 9-13 to figures 10-14. |
It will be nice to have a table with data from the devices used in the prototype. |
Thank you for your remark. To not add another table, please see the information in Table 3. There is now a sentence in the hardware section explaining the prototype devices are selected according to Table 3 for clarification. |
Could you go deeper in the statement of the efficiency improvement? |
Thank you for your comment. To further explain the experimental results and efficiency improvement, two sections were added to the experimental results. First, an overview of the efficiency measurement procedure, followed by a deeper explanation of the better efficiency achieved in the DVFW. |
Please, don’t repeat into the conclusions section what you have done, that’s not a conclusion |
Thank you for your remark. The conclusion part has been modified and we removed the unnecessary part. |

Reviewer 2 Report
This paper proposes a new topology for the Forward Voltage Converter, aimed to be used in the Universal Mains. The authors claim that this topology is versatile enough to operate both at the Low Line or the High Line voltage levels, without any need of selection between any of them. The authors also claim that the proposed topology can provide a lower stress on the electronic components that make up the system, regardless of using either voltage levels. Moreover, there is also an improvement in the overall efficiency. Along with the new topology at the power section of the converter, a controller circuit was devised, to ensure the best balancing of voltage and current between MOSFETs, and to address any significant unmatched feature regarding parasitic capacitance. A literature review was undertaken, to introduce the subject and to place the proposed topology in its field. As per the results shown, although needing more components, the proposed converter does not add volume nor cost, when compared to the conventional one. A physical circuit was fabricated, which greatly improves the manuscript content, given that the results were not only obtained from simulations, but also from a laboratorial prototype. The quality of the written English is very good.
However, there are some remarks regarding the content of the paper. The main issues are concerned about correcting some typo mistakes that need to be removed, in a paper with the purpose of having worldwide visibility, and the clarification of some aspects, as follows:
Mark 22 – The symbol PSU should be defined first.
Mark 26 – “present” should be replaced with “presents”. I suggest placing Fig. 1 as close as possible to where it is mentioned in the text.
Mark 70 – “enable” should be replaced with “enables”.
Mark 105 – Shouldn’t the capacitors be C1 and C3, instead of C2 and C4? Moreover, there is no C4 in any schematic. When comparing Fig. 1 to Fig. 3, the reference designators of the capacitors are swapped.
Mark 116 – “present” should be replaced with “presents”.
Mark 130 – In Fig. 5, D4 is always open, which contradicts what is stated in the text.
Regarding the duty-cycle values indicated in Table 2, from the text it is unclear how these values are obtained. Some sort of analytical procedure, or any other means, should be indicated, to show that the indicated values are indeed the optimal ones. Otherwise, there is no evidence to sustain these values.
Mark 157 – “require” should be replaced with “requires”.
Mark 172 – The first word in the sentence should be capitalized.
Mark 188 – “simularion” should be replaced with “simulation”.
Mark 211 – The word “dominant” should be removed.
In eq. (1), the rightmost VLP should have subscript “1”.
Mark 247 – One of the words, “with” or “and”, should be removed.
Mark 266 – “set” should be replaced with “sets”.
Mark 270 – “Expereimental” should be replaced with “Experimental”.
Mark 274 – “achieve” should be replaced with “achieves”.
It should be further explained what was the measurements procedure that allowed for obtaining the efficiency results. What parameters were measured and how were they measured and combined to yield the presented results?
Mark 281 – “265%” should be replaced with “265V”.
In the righthand plot of Fig. 9, the vertical label should be named “Efficiency deviation”, instead of just “Efficiency”.
Mark 304 – “through hall” should be replaced with “through whole”.
Mark 317 – “were” should be inserted between “experiment” and “captured”.
Mark 328 – “contains” should be replaced with “contain”.
Regarding the simulations, which component models were used? Were they devised by the authors, or were they obtained from some other source?
Author Response
This paper proposes a new topology for the Forward Voltage Converter, aimed to be used in the Universal Mains. The authors claim that this topology is versatile enough to operate both at the Low Line or the High Line voltage levels, without any need of selection between any of them. The authors also claim that the proposed topology can provide a lower stress on the electronic components that make up the system, regardless of using either voltage levels. Moreover, there is also an improvement in the overall efficiency. Along with the new topology at the power section of the converter, a controller circuit was devised, to ensure the best balancing of voltage and current between MOSFETs, and to address any significant unmatched feature regarding parasitic capacitance. A literature review was undertaken, to introduce the subject and to place the proposed topology in its field. As per the results shown, although needing more components, the proposed converter does not add volume nor cost, when compared to the conventional one. A physical circuit was fabricated, which greatly improves the manuscript content, given that the results were not only obtained from simulations, but also from a laboratorial prototype. The quality of the written English is very good. However, there are some remarks regarding the content of the paper. The main issues are concerned about correcting some typo mistakes that need to be removed, in a paper with the purpose of having worldwide visibility, and the clarification of some aspects, as follows: |
Thank you for the response. To improve the literature review, a short introduction to isolated and non-isolated topologies including new references. Furthermore, all the typos remarks have been fixed as well as a professional English review. Thank you for all your valuable feedback. |
Mark 22 – The symbol PSU should be defined first. |
Thank you for your suggestion. Added an abbreviated section for easier reading. We have checked the whole document to make sure all abbreviations are defined. |
Mark 26 – “present” should be replaced with “presents”. |
Thank you for your remark. This error has been fixed. |
I suggest placing Fig. 1 as close as possible to where it is mentioned in the text. |
Thank you for your comment. To address this issue, Figure 1 moved to be placed closer to the reference in the text.
|
Mark 70 – “enable” should be replaced with “enables”. |
Thank you for your remark. This error has been fixed. |
Mark 105 – Shouldn’t the capacitors be C1 and C3, instead of C2 and C4? Moreover, there is no C4 in any schematic. When comparing Fig. 1 to Fig. 3, the reference designators of the capacitors are swapped. |
Thank you for your comment. You are correct and it should be C1 and C3 instead of C2 and C4. Also, Fig. 3 capacitors were swapped and fixed. |
Mark 116 – “present” should be replaced with “presents”. |
Thank you for your remark. This error has been fixed. |
Mark 130 – In Fig. 5, D4 is always open, which contradicts what is stated in the text. |
Thank you for your comment. D4 is always open (OFF) in Fig 5 because the state switches are ON. Therefore, the input diode cathode has a potential of ~Vin voltage (the bulk capacitor rectified positive voltage), the Anode has a potential of ~0V (the bulk capacitor rectified ground voltage) and the diode experience negative voltage (-Vin). Hence the diode is in blocking mode. Note that the voltage is not exactly Vin and 0V as there is a small voltage drop on the State switches. |
Regarding the duty-cycle values indicated in Table 2, from the text it is unclear how these values are obtained. Some sort of analytical procedure, or any other means, should be indicated, to show that the indicated values are indeed the optimal ones. Otherwise, there is no evidence to sustain these values. |
Thank you for your comment. A clarification sentence was added just before Table 2, which explains how the values were calculated and what is the difference between the conventional Forward converter and the DVFW at the low line and high line. |
Mark 157 – “require” should be replaced with “requires”. |
Thank you for your remark. This error has been fixed. |
Mark 172 – The first word in the sentence should be capitalized. |
Thank you for your remark. This error has been fixed and other non-capitalized letters have been searched in the article. |
Mark 188 – “simularion” should be replaced with “simulation”. |
Thank you for your remark. This error has been fixed and other typos have been searched in the article. |
Mark 211 – The word “dominant” should be removed. |
Thank you for your remark. This error has been fixed. |
In eq. (1), the rightmost VLP should have subscript “1”. |
Thank you for your remark. This error has been fixed. |
Mark 247 – One of the words, “with” or “and”, should be removed. |
Thank you for your remark. This error has been fixed. |
Mark 266 – “set” should be replaced with “sets”. |
Thank you for your remark. This error has been fixed. |
Mark 270 – “Expereimental” should be replaced with “Experimental”. |
Thank you for your remark. This error has been fixed and other typos have been searched in the article. |
Mark 274 – “achieve” should be replaced with “achieves”. |
Thank you for your remark. This error has been fixed. |
It should be further explained what was the measurements procedure that allowed for obtaining the efficiency results. What parameters were measured and how were they measured and combined to yield the presented results? |
Thank you for your comment. To better explain the efficiency measurement procedure, a short paragraph followed by a new figure (Fig. 9) added to the article. |
Mark 281 – “265%” should be replaced with “265V”. |
Thank you for your remark. This error has been fixed and other typos have been searched in the article. |
In the righthand plot of Fig. 9, the vertical label should be named “Efficiency deviation”, instead of just “Efficiency”. |
Thank you for your remark. This error has been fixed. |
Mark 304 – “through hall” should be replaced with “through whole”. |
Thank you for your remark. This error has been fixed. |
Mark 317 – “were” should be inserted between “experiment” and “captured”. |
Thank you for your remark. This error has been fixed. |
Mark 328 – “contains” should be replaced with “contain”. |
Thank you for your remark. This error has been fixed. |
Regarding the simulations, which component models were used? Were they devised by the authors, or were they obtained from some other source? |
Thank you for your comment. A short sentence added in the simulation section explaining how the devices devised. |

Reviewer 3 Report
- Add the different waveforms of the components of the proposed topology like key waveforms.
- Discuss the modulation technique.
- Table 2, give the reference for the conventional converter.
- How authors can justify the content of table 3. What about the voltage, current, turn ON, turn OFF, the delay time of different components, parasitic elements of the components, as they higher affects the price of the components.
- Show the complete experimental setup diagram.
- Author needs to discuss the non-isolated topologies also like A high gain noninverting DC–DC converter with low voltage stress for industrial applications. This will improve the introduction.
- Fig. 10 and 11 can be improved.
- Rest is fine.
Author Response
Add the different waveforms of the components of the proposed topology like key waveforms. |
Thank you for your remark. Figures 6,7,9,10,11,12,13 changed to have no magenta, yellow or light green so it would be more readable for the reader. Furthermore, the text increased in the figures it was too small to read for better visibility. Note that a new figure was added to the article (Fig. 9) which pushed figures 9-13 to figures 10-14 |
Discuss the modulation technique. |
Thank you for your suggestion. Added in the proposed topology and operation section a few sentences which explain how the DVFW can operate with an off-the-shelf modulator. Hence, the DVFW need a similar PWM modulation as a conventional Forward converter. This remark is very important as it is one of the DVFW attractive features. |
Table 2, give the reference for the conventional converter. |
Thank you for your comment. A clarification sentence was added just before Table 2, which explains how the values were calculated and what is the difference between the conventional Forward converter and the DVFW at the low line and high line. |
How authors can justify the content of table 3. What about the voltage, current, turn ON, turn OFF, the delay time of different components, parasitic elements of the components, as they higher affects the price of the components. |
Thank you for your comment. This comparison has a few parameters which make it very difficult to compare the components. To make it relatable, the same product series devices and the same package were selected, as mentioned in the article. Furthermore, it is possible to further optimize the selection and take different product series or packages, which use other technologies. This is also mentioned in the article. The switching turn ON, turn OFF, the delay time of different components, parasitic elements of the component are relatively similar and now added to the table. Note that the parasitic capacitance of the lower voltage MOSFET is lower as expected which support lower switching losses in the DVFW. It was possible to take a 400V MOSFET for the DVFW, which would be cheaper and better performance but because it is different series (different technology) we decided to take a less optimized device but more fair comparison. |
Show the complete experimental setup diagram. |
Thank you for your comment. To better explain the efficiency measurement procedure, a short paragraph followed by a new figure (Fig. 9) added to the article. |
Author needs to discuss the non-isolated topologies also like A high gain noninverting DC–DC converter with low voltage stress for industrial applications. This will improve the introduction. |
Thank you for your comment. To improve the literature review, a short introduction to isolated and non-isolated topologies including new references.
|
Fig. 10 and 11 can be improved.
|
Thank you for your remark. Figures 6,7,9,10,11,12,13 changed to have no magenta, yellow or light green so it would be more readable for the reader. Furthermore, the text increased in the figures it was too small to read for better visibility. Note that a new figure was added to the article (Fig. 9) which pushed figures 9-13 to figures 10-14 |
Rest is fine. |
Thank you for all your valuable feedback. |
